# PeerJ

# An elaborate data set on human gait and the effect of mechanical perturbations

Jason K. Moore, Sandra K. Hnat and Antonie J. van den Bogert

Department of Mechanical Engineering, Cleveland State University, Cleveland, OH, USA

## ABSTRACT

Here we share a rich gait data set collected from fifteen subjects walking at three speeds on an instrumented treadmill. Each trial consists of 120 s of normal walking and 480 s of walking while being longitudinally perturbed during each stance phase with pseudo-random fluctuations in the speed of the treadmill belt. A total of approximately 1.5 h of normal walking ($>$5000 gait cycles) and 6 h of perturbed walking ($>$20,000 gait cycles) is included in the data set. We provide full body marker trajectories and ground reaction loads in addition to a presentation of processed data that includes gait events, 2D joint angles, angular rates, and joint torques along with the open source software used for the computations. The protocol is described in detail and supported with additional elaborate meta data for each trial. This data can likely be useful for validating or generating mathematical models that are capable of simulating normal periodic gait and non-periodic, perturbed gaits.

## INTRODUCTION

The collection of dynamical data during human walking has a long history beginning with the first motion pictures and now with modern marker based motion capture techniques and high fidelity ground reaction load measurements. Even though years of data on thousands of subjects now exist, this data is not widely disseminated, well organized, nor available with few or no restrictions. David Winter's published normative gait data (*Winter, 1990*) is widely used in biomechanical studies, yet it comes from few subjects and only a small number of gait cycles per subject. This small source has successfully enabled many other studies, such as powered prosthetic control design (*Sup, Bohara & Goldfarb, 2008*) but success in other research fields using large sets of data for discovery lead us to believe that more elaborate data sets may benefit the field of human motion studies. To enable such work, biomechanical data needs to be shared extensively, organized, and curated to enable future analysts.

There are some notable gait data sets and databases besides Winter's authoritative set that are publicly available. The International Society of Biomechanics has maintained a web page (http://isbweb.org/data) since approximately 1995 that includes data sets for download and mostly unencumbered use. For example, the kinematic and force plate measurements from several subjects from *Vaughan, Davis & O'Connor (1992)* is available on the site. At another website, the CGA Normative Gait Database, *Kirtley (2014)* curates

Corresponding author
Jason K. Moore,
moorepants@gmail.com

and shares normative clinical gait data collected from multiple labs and these datasets have influenced other studies, for example *Van den Bogert (2003)* made use of the average gait cycles from the child subjects.

*Chester, Tingley & Biden (2007)* report on a large gait database comparison where one database contained kinematic data of 409 gait cycles of children from 1 to 7 years old but the data does not seem to be publicly available. This is unfortunately typical. *Tirosh, Baker & McGinley (2010)* recognized the need for a comprehensive data base for clinical gait data and created the Gaitabase. This database may contain a substantial amount of data but it is encumbered by a complicated and restrictive license and sharing scheme. *Yun et al. (2014)* provides lower body kinematic data of single gait cycles from over one hundred subjects extracted from the large KIST Human Gait Pattern Data database which may also include a substantial amount of raw data but it is private. However, there are examples of data with less restrictions. The University of Wisconsin at LaCrosse has an easily accessible normative gait data set (*Willson & Kernozek, 2014*) from 25 subjects with lower extremity marker data from multiple gait cycles and force plate measurements from a single gait cycle. The CMU Graphics Lab Motion Capture Database (*Hodgins, 2015*) is also a good example and contains full body marker kinematics for a fair number of trials with small number of gait cycles during both walking and running.

More recent examples of biomechanists sharing their data alongside publications are: *Van den Bogert et al. (2013)* which includes full body joint kinematics and kinetics from eleven subjects walking on an instrumented treadmill and *Wang & Srinivasan (2014)* who include a larger set of data from ten subjects walking for five minutes each at three different speeds but only a small set of lower extremity markers are present. The second is notable because they publish the data in Dryad, a modern citable data repository. It is also worth noting purely visual data collections of gait, like the one presented in *Makihara et al. (2012)*, which contain videos of subjects walking on a treadmill in full clothing. This database is also unfortunately tightly secured with an extensive release agreement for reuse.

The amount of publicly available gait data is small compared to the number of gait studies that have been performed over the years. The data that is available generally suffers from limitations such as few subjects, few gait cycles, few markers, highly clinical, no raw data, limited force plate measurements, lack of meta data, non-standard formats, and restrictive licensing. To help with this situation we are making the data we collected for our research purposes publicly available and free of the previously mentioned deficiencies. Not only do we provide a larger set of normative gait data that has been previously available, we also include an even larger set of data in which the subject is being perturbed, something that does not currently exist. We believe both of these sets of data can serve a variety of use cases and hope that we can save time and effort for future researchers by sharing it.

But our reasons are not entirely altruistic, as governments and granting agencies are also encouraging researchers to share data with recent policy changes. For example, the *European Commission (2012)* has outlined publicly funded data's role in innovation and the *White House (2013)* laid out a plan for public access to publications and data in 2013. The National Science Foundation, which partially funds this work, was ahead of the White

House and required all grants to include a data management plan in 2011. This work is a partial fulfillment of the grant requirements laid out in our grant's data management plan and we hope that this work can be a good model for dissemination of biomechanical data.

Our use case for the data is centered around the need for bio-inspired control systems in emerging powered prosthetics and orthotics. Ideally, a powered prosthetic would behave in such a way that the user would feel like their limb was never disabled. There are a variety of approaches to developing bio-inspired control systems, some of which aim to mimic the reactions and motion of an able-bodied person. A modern gait lab is able to collect a variety of kinematic, kinetic, and physiological data from humans during gait. This data can potentially be used to drive the design of the human-mimicking controller. With a rich enough data set, one may be able to identify control mechanisms used during a human's natural gait and recovery from perturbations. We hypothesize that by forcing the human to recover from external perturbations, the resulting reactive actions can be used along with system identification techniques to expose the feedback related relationships among the human's sensors and actuators. With this in mind, we have collected data that is richer than previous gait data sets and may be rich enough for control identification. The data can also be used for verification purposes for controllers that have been designed in other manners, such as those constructed from first principles (e.g., *Geyer & Herr, 2010*).

With all of this in mind, we collected over seven and a half hours of gait data from fifteen able bodied subjects which amounts to over 25,000 gait cycles (*Moore, Hnat & Van den Bogert, 2014*). The subjects walked at three different speeds on an instrumented treadmill while we collected full body marker locations and ground reaction loads from a pair of force plates. The final protocol for the majority of the trials included two minutes of normal walking and eight minutes of walking under the influence of pseudo-random belt speed fluctuations. The data has been organized complete with rich meta data and made available in the most unrestrictive form for other research uses following modern best practices in data sharing (*White et al., 2013*).

Furthermore, we include a small Apache licensed open source software library for basic gait analysis and demonstrate its use in the paper. The combination of the open data and open software allow the results presented within to be computationally reproducible and instructions are included in the associated repository (https://github.com/csu-hmc/perturbed-data-paper) for reproducing the results.

## METHODS

In this section, we describe our experimental design beginning with descriptions of the participants and equipment. This is then followed by the protocol details and specifics on the perturbation design.

### Participants

Fifteen able bodied subjects including four females and eleven males with an average age of $24 \pm 4$ years, height of $1.75 \pm 0.09$ m, mass of $74 \pm 13$ kg participated in the study. The study was approved by the Institutional Review Board of Cleveland State University (# 29904-VAN-HS) and written informed consent was obtained from all participants.
**Table 1 Information about the 15 study participants in order of collection date.** The subjects are divided into those that were used for the protocol pilot trials, i.e., the first three, and those used for the final protocol. The final three columns provide the trial numbers associated with each nominal treadmill speed. The measured mass is computed from the mean total vertical ground reaction force just after the calibration pose event, if possible. If the mass is reported without an accompanying standard deviation, it is the subject's self-reported mass. Additional trials found in the data set with a subject identification number 0 are trials with no subject, i.e., unloaded trials that can be used for inertial compensation purposes, and are not shown in the table. Generated by `src/subject_table.py`.

| Id | Gender | Age (yr) | Height (m) | Mass (kg) | 0.8 m/s | 1.2 m/s | 1.6 m/s |
|----|--------|----------|------------|-----------|---------|---------|---------|
| 1  | male   | 25       | 1.87       | 101       | NA      | 6, 7, 8 | NA      |
| 11 | male   | 22       | 1.85       | 80        | 9       | 10      | 11      |
| 4  | male   | 30       | 1.76       | 74        | 12, 15  | 13      | 14      |
| 7  | female | 29       | 1.72       | $64.5 \pm 0.8$ | 16 | 17      | 18      |
| 8  | male   | 20       | 1.57       | $74.9 \pm 0.9$ | 19 | 20      | 21      |
| 9  | male   | 20       | 1.69       | $67 \pm 2$     | 25 | 26      | 27      |
| 5  | male   | 23       | 1.73       | $71.2 \pm 0.9$ | 32 | 31      | 33      |
| 6  | male   | 26       | 1.77       | $86.8 \pm 0.6$ | 40 | 41      | 42      |
| 3  | female | 32       | 1.62       | $54 \pm 2$     | 46 | 47      | 48      |
| 12 | male   | 22       | 1.85       | $74.2 \pm 0.5$ | 49 | 50      | 51      |
| 13 | female | 21       | 1.70       | $58 \pm 2$     | 55 | 56      | 57      |
| 10 | male   | 19       | 1.77       | $92 \pm 2$     | 61 | 62      | 63      |
| 15 | male   | 22       | 1.83       | $80.5 \pm 0.8$ | 67 | 68      | 69      |
| 17 | male   | 23       | 1.86       | $88.3 \pm 0.8$ | 73 | 74      | 75      |
| 16 | female | 28       | 1.69       | $56.2 \pm 0.6$ | 76 | 77      | 78      |

The data has been anonymized with respect to the participants' identities and a unique identification number was assigned to each subject. A selection of the meta data collected for each subject is shown in Table 1.

## Equipment

The data were collected in the Laboratory for Human Motion and Control at Cleveland State University, using the following equipment:

- A R-Mill treadmill which has dual 6 degree of freedom force plates, independent belts for each foot, along with lateral translation and pitch rotation capabilities (Forcelink, Culemborg, Netherlands).
- A 10 Osprey camera motion capture system paired with the Cortex 3.1.1.1290 software (Motion Analysis, Santa Rosa, CA, USA).
- USB-6255 data acquisition unit (National Instruments, Austin, Texas, USA).
- Four ADXL330 Triple Axis Accelerometer Breakout boards attached to the treadmill (Sparkfun, Niwot, Colorado, USA).
- D-Flow software (versions 3.16.1 to 3.16.2) and visual display system, (Motek Medical, Amsterdam, Netherlands).

The Cortex software delivers high accuracy 3D marker trajectories from the cameras along with data from the force plates and analog sensors (e.g., EMG/Accelerometer)

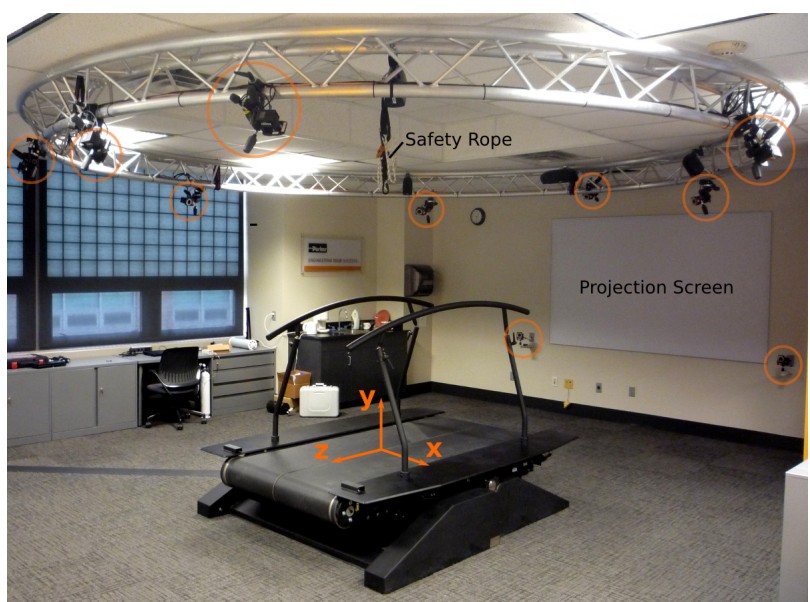

**Figure 1** **The treadmill with coordinate system, cameras (circled in orange), projection screen, and safety rope.** The direction of travel is in the $-z$ direction.

through a National Instruments USB-6255 data acquisition unit. D-Flow then receives streaming data from Cortex and any other digital sensors. It is also responsible for controlling the treadmill's motion (lateral, pitch, belts). D-Flow can process the data in real time and/or export data to file.

Our motion capture system's coordinate system is such that the X coordinate points to the right, the Y coordinate points upwards, and the Z coordinate follows from the right-hand-rule, i.e., points backwards with respect to the walking direction. The camera's coordinate system is aligned to an origin point on the treadmill's surface during camera calibration. The same point is used as the origin of the ground reaction force measuring system. Figure 1 shows the layout of the equipment.

Early on, we discovered that the factory setup of the R-Link treadmill had a vibration mode as low as 5Hz that was detectable in the force measurements; this was likely due to the flexible undercarriage and pitch motion mechanism. Trials 6–8 are affected by this vibration mode. During trials 9–15 the treadmill was stabilized with wooden blocks. During the remaining trials ($>15$) the treadmill was stabilized with metal supports; both with ones we fabricated in-house and ones supplied by the vendor. These supports aimed to improve the stiffness and frequency response of the force plate system. See the Data Limitations Section for more details.

The acceleration of the treadmill base was measured during each trial by the ADXL330 accelerometers placed at the four corners of the machine. These accelerometers were intended to provide information for inertial compensation purposes when the treadmill moved laterally or in pitch, but are extraneous for trials greater than number 8 due to the treadmill being stabilized in those degrees of freedom by the aforementioned supports.

## Protocol

The experimental protocol consisted of both static measurements and walking on the treadmill for 10 min under unperturbed and perturbed conditions. Before a set of trials on the same day, the motion capture system was calibrated using the manufacturer's recommended procedure. Before each subject's gait data were collected, the subject changed into athletic shoes, shorts, a sports bra, a baseball cap,[1] and a rock climbing harness. All 47 markers were applied directly to the skin at the landmarks noted in Table 2 except for the heel, toe, and head markers, which were placed on the respective article of clothing.[2] Then the subject self-reported their age, gender, and mass. Finally, their height was measured by the experimentalist and four reference photographs (front, back, right, left) were taken of subject's marker locations.

After obtaining informed consent and a briefing by the experimentalist on the trial protocol, the subject followed the verbal instructions of the experimentalist and the on-screen instructions from the video display. The final protocol for a single trial was as follows:

1. The subject stepped onto the treadmill and markers were identified with Cortex.
2. The safety rope was attached loosely to the rock climbing harness such that no forces were acting on the subject during walking, but so that the harness would prevent a full fall.
3. The subject started by stepping on sides of treadmill so that feet did not touch the force plates and the force plate signals are zeroed. This corresponds to the "Force Plate Zeroing" event.
4. Once notified by the video display, the subject stood in the calibration pose: standing straight up, looking forward, arms out by their sides (approximately 45 degree abduction) and the event, "Calibration Pose," was manually recorded by the operator.
5. A countdown to the first normal walking phase was displayed. At the end of the countdown the event "First Normal Walking" was recorded and the treadmill ramped up to the specified speed and the subject was instructed to walk normally, to focus on the "endless" road on the display, and not to look at their feet.
6. After 1 min of normal walking, the longitudinal perturbation phase begun and was recorded as "Longitudinal Perturbation."
7. After 8 min of walking under the influence of the perturbations, the second normal walking phase begun and was recorded as "Second Normal Walking."
8. After 1 min of normal walking, a countdown was shown on the display and the treadmill decelerated to a stop.
9. The subject was instructed to step off of the force plates for 10 s and the "Unloaded End" event was recorded.
10. The subject could then take a rest break before each additional trial.

### *Pilot protocols*

Trials 3–15 were pilot tests for finalizing the protocol design an thus have some slight variations with respect to the subsequent trials. We include these trials due to the

[1] A cap was used to avoid having to shave participants' hair at the expense of accuracy.

[2] The sacrum and rear pelvic markers were placed on the shorts for a small number of the subjects.

Table 2 **Descriptions of the 47 subject markers used in this study.** The "Set" column indicates whether the marker exists in the lower and/or full body marker set. The label column matches the column headers in the `mocap-xxx.txt` files and/or the marker map in the `meta-xxx.yml` file.

| Set | # | Label | Name | Description |
|-----|---|-------|------|-------------|
| F | 1 | LHEAD | Left head | Just above the ear, in the middle. |
| F | 2 | THEAD | Top head | On top of the head, in line with the LHEAD and RHEAD. |
| F | 3 | RHEAD | Right head | Just above the ear, in the middle. |
| F | 4 | FHEAD | Forehead | Between line LHEAD/RHEAD and THEAD a bit right from center. |
| L/F | 5 | C7 | C7 | On the 7th cervical vertebrae. |
| L/F | 6 | T10 | T10 | On the 10th thoracic vertbrae. |
| L/F | 7 | SACR | Sacrum bone | On the sacral bone. |
| L/F | 8 | NAVE | Navel | On the navel. |
| L/F | 9 | XYPH | Xiphoid process | Xiphoid process of the sternum. |
| F | 10 | STRN | Sternum | On the jugular notch of the sternum. |
| F | 11 | BBAC | Scapula | On the inferior angle of the right scapular. |
| F | 12 | LSHO | Left shoulder | Left acromion. |
| F | 13 | LDELT | Left deltoid muscle | Apex of the deltoid muscle. |
| F | 14 | LLEE | Left lateral elbow | Left lateral epicondyle of the elbow. |
| F | 15 | LMEE | Left medial elbow | Left medial epicondyle of the elbow. |
| F | 16 | LFRM | Left forearm | On 2/3 on the line between the LLEE and LMW. |
| F | 17 | LMW | Left medial wrist | On styloid process radius, thumb side. |
| F | 18 | LLW | Left lateral wrist | On styloid process ulna, pinky side. |
| F | 19 | LFIN | Left fingers | Center of the hand. Caput metatarsal 3. |
| F | 20 | RSHO | Right shoulder | Right acromion. |
| F | 21 | RDELT | Right deltoid muscle | Apex of deltoid muscle. |
| F | 22 | RLEE | Right lateral elbow | Right lateral epicondyle of the elbow. |
| F | 23 | RMEE | Right medial elbow | Right medial epicondyle of the elbow. |
| F | 24 | RFRM | Right forearm | On 1/3 on the line between the RLEE and RMW. |
| F | 25 | RMW | Right medial wrist | On styloid process radius, thumb side. |
| F | 26 | RLW | Right lateral wrist | On styloid process ulna, pinky side. |
| F | 27 | RFIN | Right fingers | Center of the hand. Caput metatarsal 3. |
| L/F | 28 | LASIS | Pelvic bone left front | Left anterior superior iliac spine. |
| L/F | 29 | RASIS | Pelvic bone right front | Right anterior superior iliac spine. |
| L/F | 30 | LPSIS | Pelvic bone left back | Left posterior superio iliac spine. |
| L/F | 31 | RPSIS | Pelvic bone right back | Right posterior superior iliac spine. |
| L/F | 32 | LGTRO | Left greater trochanter of the femur | On the cetner of the left greater trochanter. |
| L/F | 33 | FLTHI | Left thigh | On 1/3 on the line between the LFTRO and LLEK. |
| L/F | 34 | LLEK | Left lateral epicondyle of the knee | On the lateral side of the joint axis. |
| L/F | 35 | LATI | Left anterior of the tibia | On 2/3 on the line between the LLEK and LLM. |
| L/F | 36 | LLM | Left lateral malleoulus of the ankle | The center of the heel at the same height as the toe. |
| L/F | 37 | LHEE | Left heel | Center of the heel at the same height as the toe. |
| L/F | 38 | LTOE | Left toe | Tip of big toe. |
| L/F | 39 | LMT5 | Left 5th metatarsal | Caput of the 5th metatarsal bone, on joint line midfoot/toes. |
| L/F | 40 | RGTRO | Right greater trochanter of the femur | On the cetner of the right greater trochanter. |
| L/F | 41 | FRTHI | Right thigh | On 2/3 on the line between the RFTRO and RLEK. |
| L/F | 42 | RLEK | Right lateral epicondyle of the knee | On the lateral side of the joint axis. |

Table 2 (*continued*)

| Set | # | Label | Name | Description |
|-----|----|-------|------|-------------|
| L/F | 43 | RATI | Right anterior of the tibia | On 1/3 on the line between the RLEK and RLM. |
| L/F | 44 | RLM | Right lateral malleoulus of the ankle | The center of the heel at the same height as the toe. |
| L/F | 45 | RHEE | Right heel | Center of the heel at the same height as the toe. |
| L/F | 46 | RTOE | Right toe | Tip of big toe. |
| L/F | 47 | RMT5 | Right 5th metatarsal | Caput of the 5th metatarsal bone, on joint line midfoot/toes. |

uniqueness of trials 6–8 and the fact that the kinematic data is valid. We believe there may be useful analyses that only require the kinematic data. Additional information needed to interpret the data in the pilot trials can be found in the associated meta data files and the Data Limitations Section of this paper.

Trials 3–8 use an early experimental protocol which divided the walking period into three sections: no perturbation, longitudinal perturbation, and a combination of longitudinal and lateral perturbation. The calibration pose and zeroing events are present in the data but lumped into one event. These trials only use the lower body marker set described in Table 2. Additionally, there are five markers that have labels beginning with ROT that were attached to the treadmill base to capture the lateral motion. Trials 9–15 use the final protocol but have corrupt ground reaction loads due to the wooden treadmill base stabilizers.

## Perturbation signals

As previously described, the protocol included a phase of normal walking, followed by longitudinal belt speed perturbations, and ended with a second segment of normal walking. Three pseudo-random belt speed control signals, with mean velocities of $0.8 \, \mathrm{m \, s^{-1}}$, $1.2 \, \mathrm{m \, s^{-1}}$ and $1.6 \, \mathrm{m \, s^{-1}}$, were pre-generated with MATLAB and Simulink (Mathworks, Natick, Massachusetts, USA) and are available for download from Zenodo (*Hnat, Moore & Van den Bogert, 2015*). The same control signal was used for all trials at that given speed.

To create the signals, we started by generating random acceleration signals, sampled at 100 Hz, using the Simulink discrete-time Gaussian white noise block followed by a saturation block set at the maximum belt acceleration of $15 \, \mathrm{m \, s^{-2}}$. The signal was then integrated to obtain belt speed and high-pass filtered with a second-order Butterworth filter to eliminate drift. One of the three mean speeds were then added to the signal and limited between $0 \, \mathrm{m \, s^{-1}}$ to $3.6 \, \mathrm{m \, s^{-1}}$. The cutoff frequencies of the high-pass filter, as well as the variance in the acceleration signal, were manually adjusted until acceptable standard deviations for each mean speed were obtained: $0.06 \, \mathrm{m \, s^{-1}}$, $0.12 \, \mathrm{m \, s^{-1}}$ and $0.21 \, \mathrm{m \, s^{-1}}$ for the three speeds, respectively. These ensured that the test subjects were sufficiently perturbed at each speed, while remaining within the limits of our equipment and testing protocol. To ensure that the treadmill belts could accelerate to the desired values, the high performance mode in the D-Flow software was enabled. The MATLAB script and Simulink model produce a comma-delimited text file of with the desired belt speed signals indexed by the time stamp.

Figure 2 gives an idea of the effect of the treadmill and controller dynamics by plotting the measured speed of the treadmill belts from loaded trials (76, 77, 78) against the

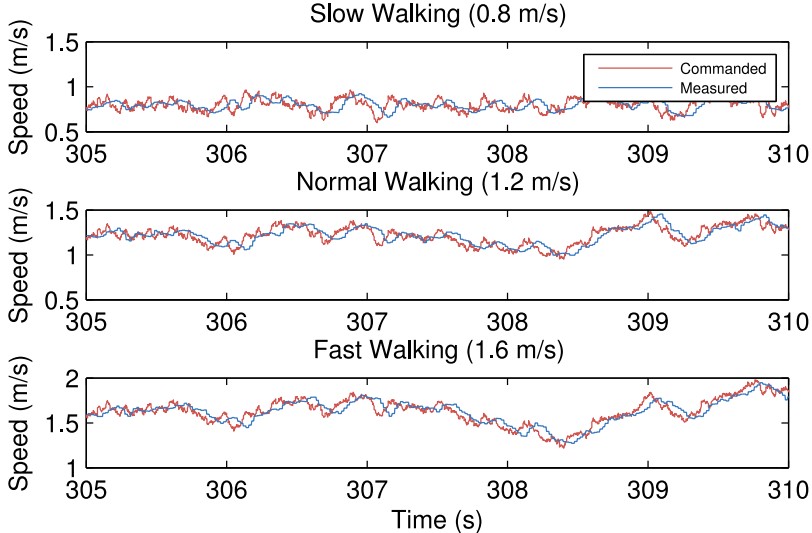

**Figure 2 Commanded treadmill belt speed (red) and the recorded speed (blue) for average belt speeds of 0.8 m s$^{-1}$, 1.2 m s$^{-1}$ and 1.6 m s$^{-1}$, respectively.** Generated by `src/input_output_plot.m`.

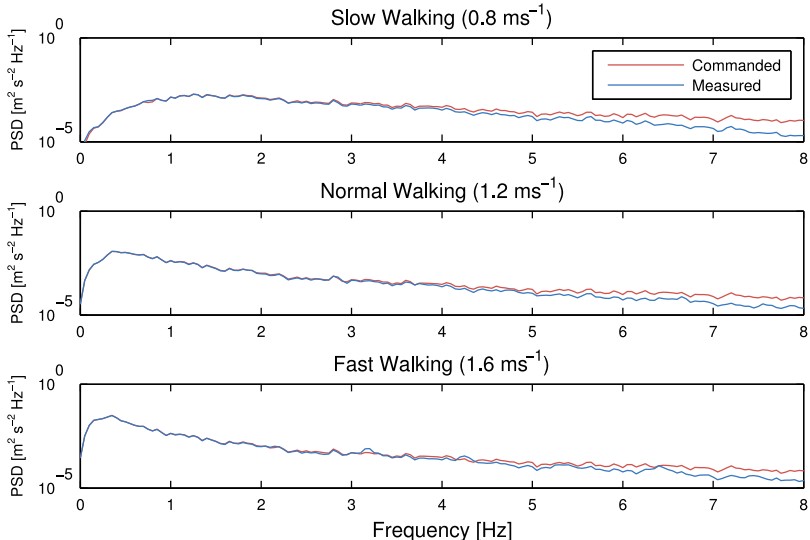

**Figure 3 Power spectral density of the commanded treadmill belt speed (red) and the recorded speed (blue) for average belt speeds of 0.8 m s$^{-1}$, 1.2 m s$^{-1}$ and 1.6 m s$^{-1}$, respectively.** Generated by `src/frequency_analysis.m`.

commanded treadmill control input signal. The system introduces a delay and seems to act as a low pass filter. The standard deviations of the measured speeds do not significantly differ from those of the commanded speeds: 0.05 m s$^{-1}$, 0.12 m s$^{-1}$ and 0.2 m s$^{-1}$ for the three speeds, respectively.

Figure 3 gives a frequency domain view of the effects of the treadmill dynamics. These spectral density plots were created by averaging a spectrogram of a twenty second Hamming window. For all speeds, the frequency content of the commanded and measured

Table 3 **A list of unloaded trials collected for each speed.** Each loaded trial includes a compensation file listed in its meta data which matches it to these unloaded trials. Generated by `src/subject_table.py`.

| Speed | Trial Numbers |
|---|---|
| 0.8 m/s | 22, 30, 34, 43, 52, 58, 64, 70, 79 |
| 1.2 m/s | 3, 4, 5, 23, 29, 35, 44, 53, 59, 65, 71, 80 |
| 1.6 m/s | 24, 28, 36, 45, 54, 60, 66, 72, 81 |

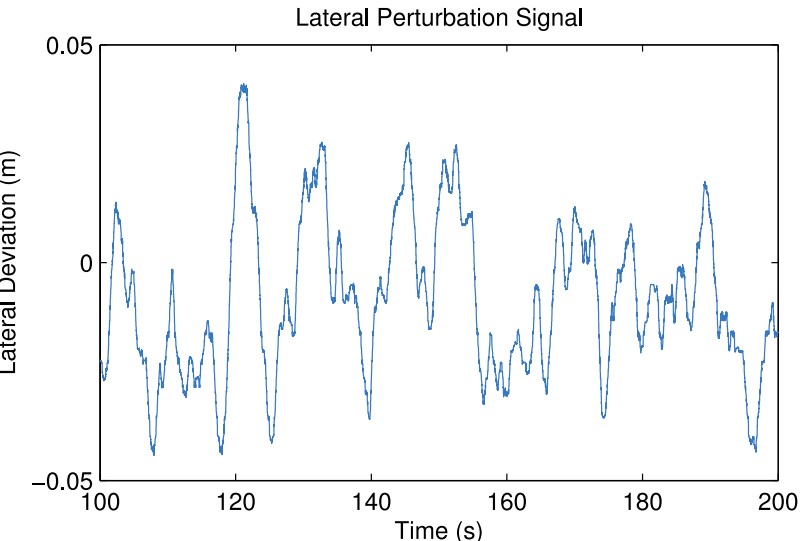

Figure 4 **The measured lateral deviation of the treadmill base from trial 6.** Generated by `src/lateral_perturbation_plot.m`.

time series show similarity below 4 Hz and attenuation in the measured spectral density above 4 Hz.

When belt speed is not constant, the inertia of the rollers and motor will likely induce error in the force plate $x$ axis moment, and hence, the anterior-posterior coordinate ($z$ axis) of the center of pressure that is measured by the instrumentation in the treadmill. This error may or may not be pertinent to different analyses. If needed, this error can be partially compensated by a linear model as shown in *Hnat & Van den Bogert (2014)*. The model coefficients can be identified from the unloaded trials given in Table 3. The error due to inertia is random and does not affect the averaged joint moments presented in Fig. 5. Compensation should, however, be done if joint moments from individual gait cycles are of interest rather than the ensemble average.

In addition to the longitudinal perturbations, lateral perturbations were also prescribed for a duration of four minutes in the pilot trials 3–8. Figure 4 shows an example of the measured lateral deviation of the treadmill base. These signals were generated in a similar manner using MATLAB and Simulink in which a Gaussian white noise block was twice integrated to obtain the lateral deviation. The signal was then high-pass filtered with a second-order Butterworth filter to eliminate drift and then saturated so that the signal

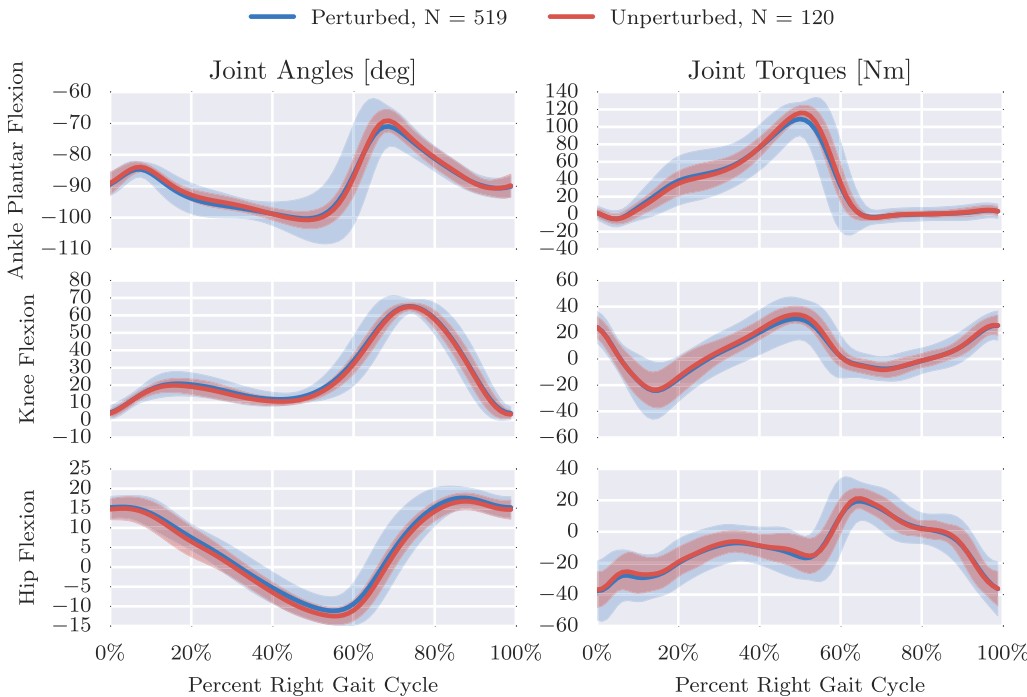

**Figure 5** **Right leg mean and 3σ (shaded) joint angles and torques from both unperturbed (red) and perturbed (blue) gait cycles from trial 20.** We define the nominal configuration, i.e., all joint angles equal to zero, such that the vectors from the shoulder to the hip, the hip to the knee, the knee to the ankle, and the heel to the toe are all aligned. Produced by `src/unperturbed_perturbed_comparison.py`.

remained within the 5 cm lateral range of the physical hardware. The same perturbation signal was used for each of the three trials.

## RESULTS

Here we present some basic results. We first provide a detailed description of the raw data followed by an overview of several computed variables that give an idea of the characteristics of both the unperturbed and perturbed gait.

### Raw data

The raw data consists of a set of ASCII tab delimited text files output from both the "mocap" and "record" modules in D-Flow in addition to a manually generated YAML[3] file that contains all of the necessary meta data for the given trial. These three files are stored in a hierarchy of directories with one trial per directory. The directories are named in the following fashion T001/ where T stands for "trial" and the following three digits provide a unique trial identification number.

### *mocap-xxx.txt*

The output from the D-Flow mocap module is stored in a tab separated value (TSV) file named `mocap-xxx.txt` where xxx represents the trial id number. The file contains a number of time series. The numerical values of the time series are provided in decimal fixed point notation with 6 decimals of precision, e.g., `123456.123456`, regardless of the

[3] YAML is a human readable data serialization format. See Listing 1 for an example.

units. The first line of the file holds the header. The header includes time stamp column, frame number column, marker position columns, force plate force/moment columns, force plate center of pressure columns, and other analog columns. The columns are further described below:

**TimeStamp** The monotonically increasing computer clock time when D-Flow receives a frame from Cortex. These are recorded approximately at 100 Hz sampling rate and given in seconds. Due to data buffering, it is preferred to derive sample times from the FrameNumber column rather than TimeStamp.

**FrameNumber** Monotonically increasing positive integers that correspond to each frame received from Cortex.

**Marker Coordinates** Any column that ends in `.PosX`, `.PosY`, or `.PosZ` are marker coordinates expressed in Cortex's Cartesian reference frame. The prefixes match the marker labels given in Table 2. These values are in meters.

**Ground Reaction Loads** There are three ground reaction forces and three ground reaction moments recorded by each of the two force plates in Newtons and Newton-Meters, respectively. The prefix for these columns is either `FP1` or `FP2` and represents either force plate 1 (left) or 2 (right). The suffixes are either `.For[XYZ]`, `.Mom[XYZ]` for the forces and moments, respectively. The force plate voltages are sampled at a much higher frequency than the cameras, but delivered at the Cortex camera sample rate, approximately 100 Hz, through the D-Flow mocap module. A force/moment calibration matrix stored in Cortex converts the voltages to forces and moments before sending it to D-Flow. The software also computes the center of pressure from the forces, moments, and force plate dimensions. These have the same prefixes for the plate number, have the suffix `.Cop[XYZ]`, and are given in meters.

**Analog Channels** Several analog signals are recorded under column headers `Channel[1-99].Anlg`. These correspond to analog signals sampled by Cortex and correspond to the 96 analog channels in the National Instruments USB-6255. The first twelve are the voltages from the force plate load cells. We also record the acceleration of 4 points on the treadmill base in analog channels 61–72 that were in place in case inertial compensation for the lateral treadmill movement was required.

We make use of the full body 47 marker set described in *Van den Bogert et al. (2013)* and presented in detail in Table 2. As with all camera based motion capture systems, the markers sometimes go missing in the recording. When a marker goes missing, if the data was recorded in a D-Flow version less than 3.16.2rc4, D-Flow continues to record the last non-missing value in all three axes until the marker is visible again. In D-Flow versions greater than or equal to 3.16.2rc4, the missing markers are indicated in the TSV file as either `0.000000` or `-0.000000`. The D-Flow version must be provided in the meta data YAML file to be able to distinguish this detail.

### record-xxx.txt

The record module also outputs a tab delimited ASCII text file with numerical values at six decimal digits. It includes a `Time` column which records the D-Flow system time in

seconds. This time corresponds to the time recorded in the `TimeStamp` column in mocap module TSV file which is necessary for time synchronization. There are two additional columns `RightBeltSpeed` and `LeftBeltSpeed` which provide the independent belt speeds measured in meters per second by a factory installed encoder in the treadmill.

Additionally, the record module is capable of recording the time at which various preprogrammed events occur, as detected or set by D-Flow. It does this by inserting commented (`#`) lines in between the rows when the event occurred. The record files have several events that delineate the different phases of the protocol:

**A: Force Plate Zeroing** Marks the time at the beginning of the trial at which there is no load on the force plates and when the force plate voltages were zeroed.

**B: Calibration Pose** Marks the time at which the person is in the calibration pose.

**C: First Normal Walking** Marks the time when the treadmill begins Phase 1: constant belt speed.

**D: Longitudinal Perturbation** Marks the time when the treadmill begins Phase 2: longitudinal perturbations in the belt speed.

**E: Second Normal Walking** Marks the time when phase 3 starts: constant belt speed.

**F: Unloaded End** Marks the time at which there is no load on the force plates and the belts are stationary.

### *meta-xxx.yml*

Each trial directory contains a meta data file in the YAML format named in the following style `meta-xxx.yml` where `xxx` is the three digit trial identification number. There are three main headings in the file: `study`, `subject`, and `trial`. An example meta data file is shown in Listing 1.

The `study` section contains identifying information for the overall study, an identification number, name, and description. This is the same for all meta data files in the study. Details are given below:

**id** An integer specifying a unique identification number of the study.
**name** A string giving the name of the study.
**description** A string with a basic description of the study.

The `subject` section provides key value pairs of information about the subject in that trial. Each subject has a unique identification number along with basic anthropomorphic data. The following details the possible meta data for the subject:

**age** An integer age in years of the subject at the time of the trial.
**ankle-width-left** A float specifying the width of the subjects left ankle.
**ankle-width-right** A float specifying the width of the subjects right ankle.
**ankle-width-units** A string giving the units of measurement of the ankle widths.
**id** An unique identification integer for the subject.
**gender** A string specifying the gender of the subject.
**height** A float specifying the measured height of the subject (with shoes and hat on) at the time of the trial.

**height-units** A string giving the units of the height measurement.

**knee-width-left** A float specifying the width of the subjects left knee.

**knee-width-right** A float specifying the width of the subjects right knee.

**knee-width-units** A string giving the units of measurement of the knee widths.

**mass** A float specifying the self-reported mass of the subject.

**mass-units** A string specifying the units of the mass measurement.

The `trial` section contains the information about the particular trial. Each trial has a unique identification number along with a variety of other information, detailed below:

**analog-channel-map** A mapping of the strings D-Flow assigns to signals emitted from the analog channels of the NI USB-6255 to names the user desires.

**cortex-version** The version of Cortex used to record the trial.

**datetime** A date formatted string giving the date of the trial in the `YYYY-MM-DD` format.

**dflow-version** The version of D-Flow used to record the trial.

**events** A key value map which prescribes names to the alphabetic events recorded in the record file.

**files** A key value mapping of files associated with this trial where the key is the D-Flow file type and the value is the path to the file relative to the meta file. The compensation file corresponds to an unloaded trial collected on the same day that could be used for inertial compensation purposes, if needed.

**hardware-settings** There are tons of settings for the hardware in both D-Flow, Cortex, and the other software in the system. This contains any non-default settings.

**high-performance** A boolean value indicating whether the D-Flow high performance setting was on (True) or off (False).

**id** An unique three digit integer identifier for the trial. All of the file names and directories associated with this trial include this number.

**marker-map** A key value map which maps marker names in the mocap file to the user's desired names for the markers.

**marker-set** Indicates the HBM (*Van den Bogert et al., 2013*) marker set used during the trial, either full, lower, or NA.

**nominal-speed** A float representing the nominal desired treadmill speed during the trial.

**nominal-speed-units** A string providing the units of the nominal speed.

**notes** A string with any notes about the trial.

**pitch** A boolean that indicates if the treadmill pitch degree of freedom was actuated during the trial.

**stationary-platform** A boolean that indicates whether the treadmill sway or pitch motion was actuated during the trial. If this flag is false, the measured ground reaction loads must be compensated for the inertial affects and be expressed in the motion capture reference frame.

**subject-id** An integer corresponding to the subject in the trial.

**sway** A boolean that indicates if the treadmill lateral degree of freedom was actuated during the trial.

```yaml
study:
    id: 1
    name: Gait Control Identification
    description: Perturb the subject during walking and running.
subject:
    id: 8
    age: 20
    mass: 70.0
    mass-units: kilograms
    height: 1.572
    height-units: meters
    knee-width-left: 107.43
    knee-width-right: 107.41
    knee-width-units: millimeters
    ankle-width-left: 70.52
    ankle-width-right: 67.66
    ankle-width-units: millimeters
    gender: male
trial:
    id: 58
    subject-id: 8
    datetime: 2014-03-28
    notes: >
        The subject did a somersault during this trial instead of following
        instructions to walk. Will have to use for another study.
    nominal-speed: 0.8
    nominal-speed-units: meters per second
    stationary-platform: True
    pitch: False
    sway: False
    hardware-settings:
        high-performance: True
    dflow-version: 3.16.1
    cortex-version: 3.1.1.1290
    marker-set: full
    marker-map:
        M1: LHEAD
        M2: THEAD
        M3: RHEAD
        M4: FHEAD
        M5: C7
    analog-channel-map:
        Channel1.Anlg: F1Y1
        Channel2.Anlg: F1Y2
        Channel3.Anlg: F1Y3
        Channel4.Anlg: F1X1
    events:
        A: Force Plate Zeroing
        B: Calibration Pose
        C: First Normal Walking
        D: Longitudinal Perturbation
        E: Second Normal Walking
        F: Unloaded End
    files:
        compensation: ../T057/mocap-057.txt
        mocap: mocap-058.txt
        record: record-058.txt
        meta: meta-058.yml
```

**Listing 1:** A fictitious example of a YAML formatted meta data file. Examples of all of the possible keys in the data set are shown.

## Processed data

We developed a toolkit for data processing, GaitAnalysisToolKit v0.1.2 (*Moore et al., 2014*) for common gait computations and provide an example processed trial to present the nature of the data. The tool was developed in Python, is dependent on the SciPy Stack [NumPy (*Walt, Colbert & Varoquaux, 2011*), SciPy (*Jones et al., 2001*), matplotlib (*Hunter, 2007*), Pandas (*McKinney, 2010*), etc] and Octave (*Octave community, 2014*), and provides two main classes: one to do basic gait data cleaning from D-Flow's output files, `DFlowData`, and a second to compute common gait variables of interest, `GaitData`.

The `DFlowData` class collects and stores all the raw data presented in the previous section and applies several "cleaning" operations to transform the data into a usable form. The cleaning process follows these steps:

1. Load the meta data file into a Python dictionary.
2. Load the D-Flow mocap module TSV file into Pandas `DataFrame`.
3. Relabel the column headers to more meaningful names if this is specified in the meta data.
4. Optionally identify the missing values in the mocap marker data and replace them with `numpy.nan`.
5. Optionally interpolate the missing marker values and replaces them with interpolated estimates using a variety of interpolation methods.
6. Load the D-Flow record module TSV file into a Pandas `DataFrame`.
7. Extract the events and create a dictionary mapping the event names in the meta data to the events detected in the record module file.
8. Inertially compensate the ground reaction loads based on whether the meta data indicates there was treadmill motion.
9. Merge the data from the mocap module and record module into one data frame at the maximum common constant sample rate.

Once the data is cleaned there are two methods that allow the user to extract the cleaned data: either extract sections of the data bounded by the events recorded in the `record-xxx.txt` file or save the cleaned data to disk. These operations are available as a command line application and as an application programming interface (API) in Python. An example of the `DFlowData` API in use is provided in Listing 2.

The `GaitData` class is then used to compute gait events (toe off and heel strike times), basic 2D inverse kinematics and dynamics, and to store the data into a Pandas `Panel` with each gait cycle on the item axis at a specified sampling rate. This object can also be serialized to disk in HDF5 format. An example of using the Python API is shown in Listing 3.

A similar work flow was used to produce Fig. 5 which compares the mean and standard deviation of sagittal plane joint angles and torques from the perturbed gait cycles to the unperturbed gait cycles computed from trial 20. This gives an idea of the more highly variable dynamics required to walk while being longitudinally perturbed.

```
>>>  from gaitanalysis.motek import DFlowData
>>>  data = DFlowData('mocap-020.txt', 'record-020.txt',
...                    'meta-020.yml')
>>>  mass = data.meta['subject']['mass']
>>>  data.clean_data()
>>>  event_df = data.extract_processed_data(
...      event='Longitudinal Perturbation')
```

**Listing 2:** Python interpreter session showing how one could load a trial into memory, extract the subject's mass from the meta data, run the data cleaning process, and finally extract a Pandas `DataFrame` containing all of the time histories for a specific event in the trial.

```
>>>  from gaitanalysis.gait import GaitData
>>>  gdata = GaitData(event_df)
>>>  gdata.inverse_dynamics_2d(left_markers, right_markers,
...                            left_loads, right_loads, mass, 6.0)
>>>  gdata.grf_landmarks('Right Fy', 'Left Fy', threshhold=20.0)
>>>  gdata.split_at('right')
>>>  gdata.plot_gait_cycles('Left Hip Joint Torque', mean=True)
>>>  gdata.save('gait-data.h5')
```

**Listing 3:** Python interpreter session showing how one could use the `GaitData` class to load in the result of `DFlowData` and compute the inverse dynamics (joint angles and torques), identify the gait events (e.g., heel strikes), split the data with respect to the gait events into a Pandas `Panel`, plot the mean and standard deviation of one time history with respect to the gait cycles, and save the data to disk.

For more insight into the difference in the unperturbed and perturbed data, Fig. 6 compares the distribution of a few gait cycle statistics. One can see that the perturbed strides have a much larger variation in frequency and length and even larger variation in stride width. It is also interesting to note that the coupled nature of the system's degrees of freedom can be exploited to increase the stride width with only longitudinal perturbations, although not relatively as much as is in the other statistics.

## Data limitations

The data is provided in good faith with great attention to detail but as with all data there are anomalies that may affect the use and interpretation of results emanating from the data. The following lists give various notes and warnings about the data that should be taken into account before use.

### All trials

- Be sure to read the notes in each meta data file for details about possible anomalies in that particular trial. Things such as marker dropout, ghost markers, and marker movement are the more prominent notes. Details about variations in the equipment on the day of the trial are also mentioned.

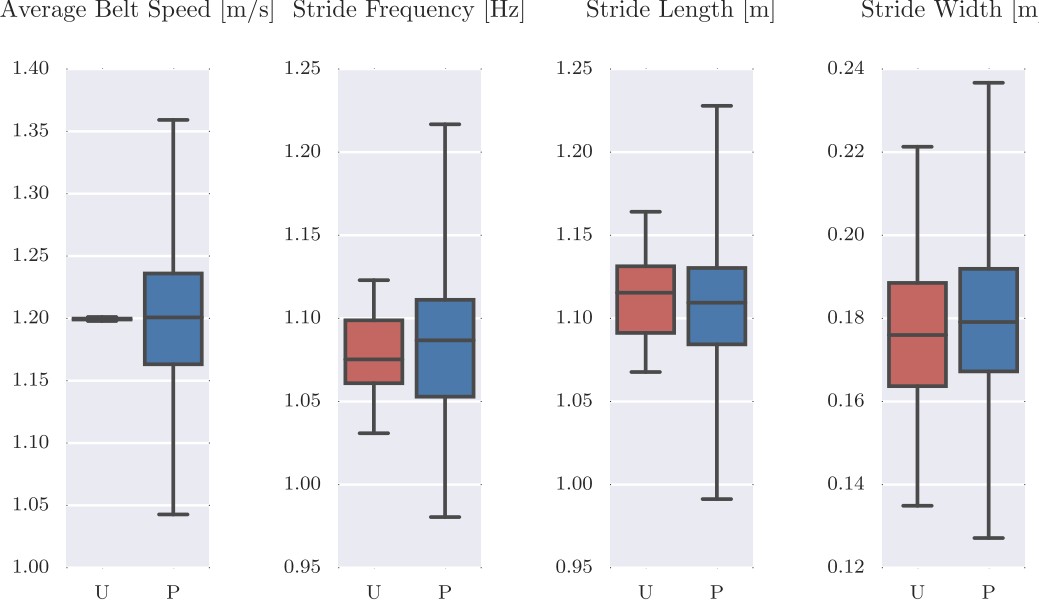

**Figure 6 Box plots of the average belt speed, stride frequency, stride length, and stride width which compare 120 unperturbed (U: red) and 519 perturbed (P: blue) gait cycles.** The median is given with the box bounding the first and third quartiles and the whiskers bound the range of the data. Produced by `src/unperturbed_perturbed_comparison.py`.

- The subject identification number 0 represents the "null subject" and was used whenever data was collected from the system with no subject on the treadmill, for example during the trials that were intended to be used for inertial compensation purposes. These trials play through the exact protocol as those with a human subject and the matching trials are indicated in the meta data. Matching unloaded trials were recorded on the same day as the loaded trials and is noted in the `trial:files:compensation` section of the meta data file. See Table 3 for a list of all the compensation trials.

- Trials 1 and 2 were not recorded as part of this study. Those trial identification numbers were reserved for early data exploration from data collected in other studies and are not included in this dataset.

- Trials 37, 38, and 39 do not exist. The numbers were accidentally skipped.

- The ankle joint torques computed from subject 9's data in trials 25–27 are abnormal and should be used with caution or not at all. We were not able to locate the source of the error, but it is likely related to the force calibration.

### Pilot trials

- Subject 1 walked only at a single speed with three trial repetitions.

- Trials 6–8 included a calibration pose at the start of the trial but the event was not explicitly recorded. In those trials, the "TreadmillPerturbation" event marks the beginning of longitudinal perturbations and the "Both" event marks the beginning of combined longitudinal and lateral perturbations. The force plate zeroing at the end was also not explicitly recorded.

- Trials 6–8's force measurements are affected by the treadmill vibration mode mentioned in the equipment section and the force plate data should not be used.
- Trials 9–11 used a slightly different event definition where the calibration poses were not explicitly tagged by an event, yet the protocol was identical to the following trials. The calibration pose will have to be determined manually.
- During trials 9–15, we used wooden blocks to fix the treadmill to the concrete floor to eliminate the treadmill's low vibration mode. But these blocks seem to have corrupted the force plate measurements by imposing frictional stresses on the system. The force plate measurements should not be used from these trials.
- We did not record unloaded compensation trials for trials 9–15. Regardless, they would likely be useless due to the corruption from the wooden blocks and are not needed because the treadmill base is fixed.

## CONCLUSION

We have presented a rich and elaborate data set of motion and ground reaction loads from human subjects during both normal walking and when recovering from perturbations. The raw data is provided for reuse with complete meta data. In addition to the data, we provide software that can process the data for both cleaning purposes and to produce typical sagittal plane gait variables of interest. Among other uses, we believe the dataset is ideally suited for control identification purposes. Many researchers are working on mathematical models for control in gait and this dataset provides both a way to validate these models and a source for generating them.

## ACKNOWLEDGEMENTS

We thank Roman Boychuk and Obinna Nwanna for assistance with the experiments. We also thank Sabrina Abram, Brad Humphreys, and Anne Koelewijn for reviewing the preprint and being our guinea pigs on the software/data instructions. Dan Simon also gave valuable feedback on the preprint. Furthermore, we thank the academic editor, Arti Ahluwalia, and three reviewers, Morgan Sangeux, Paul Lee, and Manoj Srinivasan, for their valuable feedback which helped improve the quality of the paper and data.

### Funding

The work was partially funded by the State of Ohio Third Frontier Commission through the Wright Center for Sensor Systems Engineering (WCSSE) and by the National Science Foundation under Grant No. 1344954. The funders had no role in study design, data collection and analysis, decision to publish, or preparation of the manuscript.

### Grant Disclosures

The following grant information was disclosed by the authors:
Wright Center for Sensor Systems Engineering (WCSSE).
National Science Foundation: 1344954.

## Competing Interests

The authors declare there are no competing interests.

## Author Contributions

- Jason K. Moore and Sandra K. Hnat conceived and designed the experiments, performed the experiments, analyzed the data, contributed reagents/materials/analysis tools, wrote the paper, prepared figures and/or tables, reviewed drafts of the paper.
- Antonie J. van den Bogert conceived and designed the experiments, contributed reagents/materials/analysis tools, wrote the paper, reviewed drafts of the paper.

## Human Ethics

The following information was supplied relating to ethical approvals (i.e., approving body and any reference numbers):

The study was approved by the Institutional Review Board of Cleveland State University (# 29904-VAN-HS) and informed consent was obtained from all participants.

## Data Deposition

The following information was supplied regarding the deposition of related data:

Zenodo:

– http://dx.doi.org/10.5281/zenodo.13030

– http://dx.doi.org/10.5281/zenodo.13253

– http://dx.doi.org/10.5281/zenodo.13159

– http://dx.doi.org/10.5281/zenodo.16064

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
