# Peer review of "An elaborate data set on human gait and the effect of mechanical perturbations"

_PeerJ, doi:10.7717/peerj.918_

## Round 0.1 · original submission · Minor Revisions

Please give a context and objective to the work, as noted by Reviewer 1 and 2. This could stem from the funder driven requirement to share data openly (as suggested by R.3), but I am sure the data was generated and analysed for one or more specific research questions or hypotheses. Please address R2 and R3's specific comments to authors.

·

Basic reporting

The authors of the submission wish to share a comprehensive set of data related to the analysis of normal gait. I believe the data will be valuable for the gait/biomechanics community and I appreciate the work of the authors as well as their willingness to help others. However, I do not think the proposed manuscript represent a 'unit of publication'. I suspect these data were collected for a purpose but this purpose is absent from the manuscript.

Experimental design

As mentioned in basic reporting, there is no research question and therefore I do not think it can join the scholarly literature as is. I would strongly encourage the authors to re-submit their work/data as part of a paper WITH a research question.

Validity of the findings

There are no findings since there is no question.

·

Basic reporting

The submission must adhere to all PeerJ policies.
(Should be OK)
The article must be written in English using clear and unambiguous text and must conform to professional standards of courtesy and expression.
(The language was clear and the manuscript was well written.)
The article should include sufficient introduction and background to demonstrate how the work fits into the broader field of knowledge. Relevant prior literature should be appropriately referenced.
(The introduction provided adequate background for readers to understand the objective of this manuscript.)
The structure of the submitted article should conform to one of the templates. Significant departures in structure should be made only if they significantly improve clarity or conform to a discipline-specific custom.
(The manuscript followed the standard IMaRD format.)
Figures should be relevant to the content of the article, of sufficient resolution, and appropriately described and labeled.
The submission should be ‘self-contained,’ should represent an appropriate ‘unit of publication’, and should include all results relevant to the hypothesis. Coherent bodies of work should not be inappropriately subdivided merely to increase publication count.
(I am concern about this point, as this manuscript described an open-source data of gait movement, without any data analysis or hypothesis testing. Although I agree that the data are useful and worth publication, the authors could divide the results of their study for few more papers.)

Experimental design

The submission must describe original primary research within the Scope of the journal.
(The authors mentioned that the objective of this study was to provide a rich gait movement data with fluctuations in speed, and I don't believe this was a primary research.)
The submission should clearly define the research question, which must be relevant and meaningful.
(Please see response above)
The investigation must have been conducted rigorously and to a high technical standard.
(This is OK.)
Methods should be described with sufficient information to be reproducible by another investigator.
(The authors did a good job in describing their study.)
The research must have been conducted in conformity with the prevailing ethical standards in the field.
(This study followed ethical standards in the field.)

Validity of the findings

(The authors had not described any results or data analysis. This manuscript only outlined the dataset, and I believe this section is not applicable for this manuscript.)

Additional comments

Table 1. Why subject ID 1 trialled 1.2 m/s for three times but did not trialled 0.8 m/s and 1.6 m/s?

Page 4, line 120. Why the participants were required to wear a baseball cap?

Page 15, line 374. Why didn't the authors recode the trial numbers from 40 onwards to 37, 38, ...?

·

Basic reporting

No comments.

Experimental design

No comments.

Validity of the findings

No comments.

Additional comments

This article is an important (if unconventional) addition to the biomechanics literature. The article is a wonderful example of attention to detail in presenting the protocol and experiment used, in describing formatting and reliability of the data, and in providing simple computational tools (that do not require any proprietary data) for simple processing of the data. I believe that this article will be important in the field, and I hope that other researchers will follow Moore et al’s lead in sharing and documenting their data — that this is not a one-off but something everyone does.

The following are some minor comments and suggestions:

The authors could refer to new/emerging guidelines by some funding agencies (NSF, etc.) and some journals (Royal Society journals) that insist on making available all data funded by them or published by them. And your example could be a good model for such ‘required’ publication of data.

The citation style seems a bit unorthodox, is this the Peerj recommendation? For instance,
“David Winter’s published normative gait data, Winter (1990), is widely used in biomechanical studies …”
could be:
“David Winter’s published normative gait data (Winter, 1990) is widely used in biomechanical studies …”

“At another website, the CGA Normative Gait Database, Kirtley (2014) shares normative gait data from several studies …”
Perhaps this sentence could be edited to avoid the possible misunderstanding that Kirtley conducted all these several studies. Might it be worth also citing the original studies from which the data is taken? This might be appropriate and feasible if you citations with numbers like [5-10].

Physionet (http://www.physionet.org/), the Carnegie Mellon mocap database ( http://mocap.cs.cmu.edu/ ), the Ohio State mocap database, the OU-ISIR database ( http://www.am.sanken.osaka-u.ac.jp/BiometricDB/GaitTM.html ), KIST database ( http://www.me.utexas.edu/~reneu/res/gait_toolbox.html ) are some other sources of public data of aspects of human movement, but again, all these either suffer from some of the issues that the authors point out, or mainly meant for video games, animations, or biometry rather than for detailed biomechanical analyses. Please include some such databases in your introductory discussion (ones that seem most relevant).

Line 110. “Acceleration of treadmill” . Perhaps say “acceleration of treadmill base” or something so as to distinguish from the belt accelerations?

page 6. The description of the ‘perturbation signals’ on page 6 does not explain what, if any, lateral movements of the treadmill base were imposed. Line 151 alludes to the possibility of ‘both’ longitudinal and lateral perturbations.

Are the lateral perturbations used only in trials 6-8? I did not see the ‘both’ event for treadmill perturbations in the few other trial YAML files I looked at. Perhaps make an explicit note of this.

If the perturbations are only in the fore-aft direction, it is possible that the data set is insufficiently rich to infer the human walking control system; but it is also possible that the data set is rich enough due to sufficient coupling of the various degrees of freedom (fore-aft and sideways degrees of systems, to be a bit colloquial).

Line 184-185. “When belt speed is not constant, the inertia of the rollers and motor will induce error in the force plate x axis moment, and hence, the anterior-posterior coordinate (z axis) of the center of pressure that is measured by the instrumentation in the treadmill.”
This comment by the authors creates doubt in the reader’s mind as to whether the other force values are reliable. Perhaps the authors could add an explicit note allaying any such doubts.

Line 194. The abbreviation ‘YAML’ is used without previous definition. While ASCII is a common-enough word, I’d suggest that YAML is not. Perhaps the authors could explain what YAML is in the following sentence, and then refer to one of their YAML listings (Listing 1.) in that sentence. Please look for other uncommon abbreviations to clarify throughout the paper.



232. TSV (tab separated file). perhaps the expansion in parenthesis.

Figure 4, caption. Perhaps indicate in your caption what the ‘zero’ for your angles correspond to? Alternatively, what does the ‘calibration pose’ correspond to? I believe that the convention used is different from, for instance, David Winter’s data (which is, of course, fine). I believe -90 degrees ankle corresponds to quite standing in this figure, whereas in Winter’s data, 0 degrees ankle is close to quiet standing.

374. Is there a reason why the authors did not re-arrange the trial numbers for publication — ignoring accidentally skipped trials, etc?

Figure 5. Nice figure. It would be interesting to see step width distributions as well, comparing perturbed and unperturbed, as it would answer my question of whether people’s sideways dynamics were substantially affected as well. This is not absolutely necessary for the point that the authors wish to make, but could be a quick easy thing for the authors to generate from their data (especially given that they have already estimated the stride-length, step width is probably only a couple of lines of code!).

pages 16-17. I especially liked the ‘Data Limitations’ section. But I would suggest that all of these limitations be included as part of the meta data in the corresponding YAML files. For instance, in lines 378-381, you state that the force measurements should not be trusted in trials 6-15. I checked the YAML file for trial 6 and 15 (T006 and T015) to see if its ‘notes’ contained the same note, and it did not (unless I missed something). I think this would be very useful. Of course, I do see that other types of limitations or explanations are in the ‘notes’ section of the YAML file.

---

## Round 0.2 · accepted · Accept

Please add some examples of analysis which your data could facilitate, even if out of the context of your research goals. This may help readers apply the data.

·

Basic reporting

My main objection with the paper was related to PeerJ’s recommendation regarding the reviewing process:
“The submission should be ‘self-contained,’ should represent an appropriate ‘unit of publication’, and should include all results relevant to the hypothesis. Coherent bodies of work should not be inappropriately subdivided merely to increase publication count.”
I accept the authors argument that data papers are becoming more common but I am still unsure why the authors refuse to publish these data alongside, at least, one of the analysis the data was supposed to facilitate. In particular, the authors did not put forward an argument that doing so would have a detrimental effect on the paper. In the absence of a detrimental effect, why publishing a data paper rather than a scientific paper?
Having said that, I appreciate the additional information regarding the context of the work provided in the introduction and, again, think that these data will be very valuable to the community. I have no objections on the material and methods and have no reasons to question the validity of the data proposed.

Experimental design

NA

Validity of the findings

NA

Additional comments

NA

·

Basic reporting

OK.

Experimental design

OK.

Validity of the findings

OK.

Additional comments

All comments had been address and I have no further comments.